# Molecular Properties of Human Guanylate Cyclase-Activating Protein 3 (GCAP3) and Its Possible Association with Retinitis Pigmentosa

**DOI:** 10.3390/ijms23063240

**Published:** 2022-03-17

**Authors:** Anna Avesani, Laura Bielefeld, Nicole Weisschuh, Valerio Marino, Pascale Mazzola, Katarina Stingl, Tobias B. Haack, Karl-Wilhelm Koch, Daniele Dell’Orco

**Affiliations:** 1Section of Biological Chemistry, Department of Neurosciences, Biomedicine and Movement Sciences, University of Verona, 37134 Verona, Italy; anna.avesani@univr.it (A.A.); valerio.marino@univr.it (V.M.); 2Division of Biochemistry, Department of Neuroscience, University of Oldenburg, 26111 Oldenburg, Germany; laura.bielefeld@uni-oldenburg.de (L.B.); karl.w.koch@uni-oldenburg.de (K.-W.K.); 3Institute for Ophthalmic Research, Centre for Ophthalmology, University of Tübingen, 72076 Tübingen, Germany; nicole.weisschuh@uni-tuebingen.de; 4Institute of Medical Genetics and Applied Genomics, University of Tübingen, 72076 Tübingen, Germany; pascale.mazzola@med.uni-tuebingen.de (P.M.); tobias.haack@med.uni-tuebingen.de (T.B.H.); 5University Eye Hospital, Centre for Ophthalmology, University of Tübingen, 72076 Tübingen, Germany; katarina.stingl@med.uni-tuebingen.de; 6Centre for Rare Diseases, University of Tübingen, 72076 Tübingen, Germany

**Keywords:** guanylate cyclase (guanylyl cyclase), cyclic GMP, calcium-binding proteins, phototransduction, retinitis pigmentosa, vision, retina, neurodegenerative disease, *GUCA1C*, GCAP

## Abstract

The cone-specific guanylate cyclase-activating protein 3 (GCAP3), encoded by the *GUCA1C* gene, has been shown to regulate the enzymatic activity of membrane-bound guanylate cyclases (GCs) in bovine and teleost fish photoreceptors, to an extent comparable to that of the paralog protein GCAP1. To date, the molecular mechanisms underlying GCAP3 function remain largely unexplored. In this work, we report a thorough characterization of the biochemical and biophysical properties of human GCAP3, moreover, we identified an isolated case of retinitis pigmentosa, in which a patient carried the c.301G > C mutation in *GUCA1C*, resulting in the substitution of a highly conserved aspartate residue by a histidine (p.(D101H)). We found that myristoylated GCAP3 can activate GC1 with a similar Ca^2+^-dependent profile, but significantly less efficiently than GCAP1. The non-myristoylated form did not induce appreciable regulation of GC1, nor did the p.D101H variant. GCAP3 forms dimers under physiological conditions, but at odds with its paralogs, it tends to form temperature-dependent aggregates driven by hydrophobic interactions. The peculiar properties of GCAP3 were confirmed by 2 µs molecular dynamics simulations, which for the p.D101H variant highlighted a very high structural flexibility and a clear tendency to lose the binding of a Ca^2+^ ion to EF3. Overall, our data show that GCAP3 has unusual biochemical properties, which make the protein significantly different from GCAP1 and GCAP2. Moreover, the newly identified point mutation resulting in a substantially unfunctional protein could trigger retinitis pigmentosa through a currently unknown mechanism.

## 1. Introduction

Guanylate cyclase-activating proteins (GCAPs) are neuronal calcium sensors (NCS) involved in the regulation of the membrane-bound guanylate cyclases via a Ca^2+^-mediated feedback mechanism, which renders the enzymatic activity dependent on intracellular [Ca^2+^] [1]. This mechanism is essential to achieve a fine modulation of the phototransduction cascade in photoreceptors: the light-induced activation of the 3′,5′-cyclic GMP (cGMP) phosphodiesterase 6 (PDE) causes the transient closure of cyclic nucleotide-gated channels (CNG), thus provoking subtle changes in the intracellular Ca^2+^ levels, which reach low nanomolar values in conditions of light-activated photoreceptors. These changes are promptly detected by GCAPs, which respond by switching to a conformation that stimulates the GC-mediated production of cGMP in the low intracellular [Ca^2+^] phase [2]. When sufficient levels of cGMP have been replenished by the activation of GC, the CNG channels reopen, Ca^2+^ levels return to the sub-micromolar values typical of dark-adapted cells and GCAPs then act as GC inhibitors, bringing the catalytic activity down to or below the basal levels [3]. GCAPs are therefore essential players in shaping the photo response of rods and cones.

In human photoreceptors, three isoforms of GCAPs were identified, namely GCAP1, GCAP2 and GCAP3 [4], which share a high degree of sequence identity. Three-dimensional structures have been determined for all GCAPs in specific signaling states [5,6,7] and suggest high structural similarity among the homologs. The *GUCA1C* gene, encoding human GCAP3, is not expressed in bovine and mouse retinas [8,9], but immunostaining localized the protein in human cone outer segments [9] and in zebrafish cones’ outer and inner segments, but also in the outer plexiform layer [10]. Reconstitution of purified recombinant zGCAP3 with zebrafish GCs in native retina membranes showed zGCAP3 being an efficient regulator of GC activity [11] exhibiting Ca^2+^-sensing and Ca^2+^-binding in the sub-micromolar concentration range [10,12]. According to these previous investigations, humans’ and zebrafish’ GCAP3 are cone-specific proteins, the human variant sharing 45% sequence identity (approximately 70% similarity) with GCAP1, and a similar profile of GC activation and regulation in experiments performed with washed rod-outer segments from bovine retinas [8,9].

Like the other two isoforms, GCAP3 presents the typical structural features of an EF-hand Ca^2+^-binding protein, with two globular domains (N- and C-terminal), each made of a couple of EF-hand motifs flanked to each other (Figure 1). The three functional EF-hands, EF2, EF3 and EF4, permit the binding of up to three Ca^2+^ or Mg^2+^ cations. Detailed biophysical and biochemical information on the specific binding process of Mg^2+^ and Ca^2+^ binding to GCAP1 and GCAP2 is available [13,14,15,16,17] and the implications for triggering allosteric mechanisms essential to their function have been analyzed [18], however, much less information is currently available about GCAP3.

To date, both GCAP1 and GCAP2 have been associated with retinal degeneration. Indeed, more than twenty mutations in the gene *GUCA1A* encoding for GCAP1 have been associated with autosomal dominant cones or cone-rod dystrophies (CORD) [19,20,21,22,23,24,25,26,27,28,29,30,31,32], while a relatively rare missense mutation found in *GUCA1B*, encoding for GCAP2, has been associated with retinitis pigmentosa (RP), in some cases with macular involvement [33,34]. Very recently, a study reported an association between GCAP3 and autosomal recessive primary congenital glaucoma [35], although the pathogenicity of the detected variant has been questioned [36].

To date, no comprehensive biochemical and biophysical investigation on the molecular properties of human GCAP3 has been reported, nor has any association with retinal disease. In this work, we present a thorough characterization of the structural and functional properties of human GCAP3 and its capability to activate GC1 (also referred to as GC-E, ROS-GC1 or RetGC1), the main isozyme of the membrane-bound guanylate cyclase, present both in cones and rods [37,38]. We focused on the role of N-terminal myristoylation, which is known to affect Ca^2+^ affinity, protein stability [39], intracellular location [40] and GC1 regulation in the case of GCAP1 [37], and to significantly affect cation-binding and oligomeric properties in the case of bovine [15] and human GCAP2 [13].

We also identified a subject carrying a point mutation in *GUCA1C* (ENST00000261047.8: c.301G > C) in heterozygous state, whose clinical phenotype is consistent with retinitis pigmentosa, thus suggesting the involvement of GCAP3 in inherited retinal disease. At the protein level, the mutation (p.(D101H)) substitutes a highly conserved, acidic Asp residue with a positively charged His residue in the high-affinity binding site EF3, which has a prominent role in the coordination of the Ca^2+^ ion binding (Figure 1). Biophysical studies were corroborated by extensive 2 μs molecular dynamics’ (MD) simulations, in which wild-type GCAP3 was compared with the novel variant in terms of protein flexibility and cation coordination. Our findings clearly show that human GCAP3 presents unique biochemical and biophysical features not shared by its paralogs and confirm that GCAP1 is the prominent activator of GC1 in human photoreceptors.

## 2. Results

### 2.1. Identification of a Novel, Putatively Pathogenic GUCA1C Variant in Heterozygosis

The patient in this study underwent diagnostic genetic testing by genome sequencing, as previously described. Methodological details have already been published [41]. Briefly, sequencing (2 × 150 bp paired-end reads) was performed on an Illumina platform (NovaSeq6000). The average coverage on target was 47×. No putative pathogenic variants were identified in any known retinal disease gene. However, extended analysis of all protein-coding genes revealed a nucleotide substitution, c.301G > C; p.(D101H), in the heterozygous state in the *GUCA1C* gene. This variant was absent from 282,912 alleles of the Genome Aggregation database (gnomAD) [42]. Based on the nucleotide substitution c.301G > C, the exchange of an evolutionarily highly conserved amino acid localized in the second calcium-binding motif of the GCAP3 protein was predicted. The corresponding amino acids in the paralogous proteins GCAP1 and GCAP2 are located at position 100 (GCAP1) and 102 (GCAP2), respectively, and are involved in Ca^2+^-sensing [43]. Mutations at amino acid position 100 in the *GUCA1A* gene (encoding GCAP1) have previously been described in patients with autosomal dominant cone dystrophy [28,44]. Hence, we hypothesized that the c.301G > C variant in *GUCA1C* which we identified in our patient could be the underlying cause of disease.

### 2.2. Clinical Phenotype Assessment by Morphological and Electrophysiological Examinations

According to the patient, she noticed tunnel vision and blurry vision at the age of 21 but presented to our eye hospital only at the age of 33. On examination, her presenting best corrected visual acuity (BCVA) was 20/32 in the right eye and 20/40 in the left eye. The anterior segment was without pathological findings except for the beginning of a posterior polar cataract. On fundus examination (Figure 2A,B,D,E), a vital optic disc, narrowed arterioles and typical bone spicula in the retinal periphery could be seen, all typical of retinitis pigmentosa. Both eyes had a mild macular edema, which is a common complication in patients with retinitis pigmentosa. Furthermore, optical coherence tomography (OCT) examination (Figure 2C,F) showed degenerated photoreceptor outer segments up to the fovea, typical of retinitis pigmentosa, as well. Kinetic perimetry showed a concentric constriction of the visual field to 5 degrees (Appendix A). Full-field electroretinography (ERG) revealed non-measurable rod and cone responses in the standard recording. The full-field dark-adapted thresholds showed pathologically elevated, cone-mediated dark-adapted thresholds.

Although the patient presented to us at an advanced stage of disease, the clinical picture was compatible with retinitis pigmentosa.

Family history was negative for either acquired or hereditary ocular diseases. Review of the family pedigree revealed no other family member in three generations with any vision problems other than ametropia. Genotyping of the patient’s mother and brother was negative for the *GUCA1C* variant. The father was already deceased, therefore a de novo status of the variant could neither be confirmed nor excluded.

### 2.3. GC1 Regulation by D101H-GCAP3

The functional role of GCAP3 was addressed by testing its ability to regulate the activity of GC1, which is the main GC isoform involved in the cGMP production in the phototransduction [3]. GC assays were carried out with membranes containing heterologously expressed human GC1 and reconstituted with purified myristoylated and non-myristoylated wild-type (WT) human GCAP3 (mWT and nmWT, respectively) and D101H variants (mD101H and nmD101H, respectively), at high or low Ca^2+^ concentration, corresponding to the levels in dark-adapted or light-activated photoreceptors, respectively.

As shown in Figure 3A, only mWT was able to regulate GC1 in a Ca^2+^-dependent manner; indeed, in the presence of nmWT, mD101H and nmD101H variants GC1 exhibited an activity comparable to that of the control in the absence of GCAP3, indicating a lack of enzyme regulation. Despite the activation of GC1 observed at a low Ca^2+^ level by mGCAP3, cGMP synthesis was significantly lower compared to that induced by the main GC1 regulator GCAP1, as assessed by the approximately 6-fold lower capability to activate GC1 under low Ca^2+^ conditions (Figure 3B).

We then assessed the Ca^2+^-dependent activation of GC1 by mWT and mD101H GCAP3 variants (Figure 3C) by quantifying cGMP synthesis at different Ca^2+^-concentrations, mimicking the physiological variations in the photoreceptor outer segment upon cell illumination. No activation induced by the mutant was observed at the tested Ca^2+^ concentrations, while the mGCAP3 displayed a half-maximal activation of GC1 (IC_50_) at 0.52 µM Ca^2+^, thus similar to the values reported in previous studies for hGCAP1 (0.26–0.59 µM [26,45].

To probe whether the low regulation of GC1 by the WT and D101H mutant was ascribable to a reduced affinity for Ca^2+^, we assessed the ability of proteins to bind Ca^2+^ by exploiting the differential electrophoretic mobility on SDS-PAGE displayed by Ca^2+^-sensor proteins upon Ca^2+^-binding [46]. GCAPs indeed show a typical mobility shift when the apo-form is compared with the Ca^2+^- or Mg^2+^-loaded forms [31,47] and these changes are significantly altered by point mutations affecting the affinity for Ca^2+^ [43]. For proteins binding Ca^2+^ with high-affinity, such as GCAPs, the Ca^2+^-bound form results in higher electrophoretic mobility, corresponding to lower apparent molecular mass (MM) [47]. Indeed, after the addition of Ca^2+^, a decrease in electrophoretic mobility (ranging from 23.6-kDa to 18-kDa) was observed only for mWT GCAP3 with respect to the apo or Mg^2+^-bound form, whereas no shift was visible for either mutant variant. In addition, the SDS-PAGE gels showed bands at different, high MM, compatible with dimeric or aggregate GCAP3, which were confirmed by western blot analysis (Appendix A).

### 2.4. Quaternary Structure as Assessed by Analytical Size Exclusion Chromatography

The structural characterization of proteins by spectroscopic techniques requires high amounts of purified protein. Despite several expression and purification cycles performed for each variant, the mD101H resulted in a very modest amount of purified protein (<1 mg/L culture medium), which was just sufficient for the enzymatic assays and the gel-shift experiments shown in Figure 3. Moreover, any attempts to concentrate the protein for spectroscopic measurements resulted in a massive loss of protein, therefore we necessarily excluded mD101H GCAP3 from the following investigations.

Analytical size-exclusion chromatography (SEC) was performed in the presence and in the absence of cations to investigate their effects on the oligomeric state of GCAP3 variants. The elution profiles of nmWT and mWT GCAP3 in the absence of ions and in the presence of Mg^2+^ (Figure 4A,B) showed almost overlapping peaks, corresponding to a MM of ~54 and ~43 kDa, respectively, compatible with the dimeric form of the proteins. Both variants displayed a shift of the elution peak upon Ca^2+^ -binding to lower MM (41.2 kDa for nmWT and 33.8 kDa for mWT, Table 1), which was still not fully compatible with the monomeric form, but rather suggested a heterogeneous population of monomers and dimers, or a deviation from an ideal spherical structure. Notably, although SEC chromatograms were recorded in triplicate after the injection of 40 µM for all GCAP3 variants, mWT and nmD101H displayed a significantly lower maximal absorbance at 280 nm and broader peaks with respect to the nmWT in all three independent runs, suggesting an aggregation process for these variants.

A broad peak was observed for nmD101H GCAP3 (Figure 4C) in the apo form, while two largely overlapping peaks were visible in the Mg^2+^-bound form; peaks became more separated in the presence of Ca^2+^. The estimation of the MM based on the calibration curve suggested that the first elution peak was compatible with high-order oligomers, with a MM ranging between 66.9 and 72.3 kDa, while the second peak, especially marked in the presence of Ca^2+^ was compatible with a dimeric assembly. Again, the apparent MM of the nmD101H dimer was found to be cation-dependent, as shown by the decrease from 46.8 kDa upon Mg^2+^-binding to 37.6 kDa in the Ca^2+^-bound form.

### 2.5. Oligomeric Properties and Aggregation Propensity of GCAP3 Variants Are Temperature-Dependent

Analytical SEC measurements highlighted an unexpected heterogeneity of GCAP3 variants in terms of the apparent MM, suggesting that each GCAP3 oligomer may assemble with different dynamics. Since the protein aggregation process may significantly depend on temperature, we performed dynamic light scattering (DLS) measurements to elucidate the aggregation propensity and the oligomeric properties of GCAP3 variants over 5 h at two different temperatures, namely 25 °C and at 37 °C.

Unfortunately, the hydrodynamic diameter of GCAP3 oligomers could not be reliably estimated in any of the tested conditions due to the aggregation phenomena and high polydispersion, which characterized all variants. Nevertheless, the monitoring of the Mean Count Rate (MCR) over time suggested a temperature-dependent aggregation process. Both nmWT and mWT GCAP3 exhibited fairly stable MCR at 25 °C independently of the presence of cations (Figure 5A,B), whereas at 37 °C, Ca^2+^ was found to induce aggregation after ~1.5 h (Figure 5C–E). The aggregation was more apparent for mGCAP3 as the maximum of the MCR profile tripled that of nmGCAP3. The profile was not monotonic, but rather showed abrupt fluctuations in both cases, being especially scattered in the case of mGCAP3. Altogether, this suggests that, independently on myristoylation, the oligomeric state of GCAP3 is somewhat unstable in the presence of Ca^2+^.

The nmD101H variant displayed larger oscillations of the MCR with respect to the WT both in the absence of ions and in the presence of Mg^2+^, with similar trends (Figure 5C,E). In the presence of Ca^2+^, though, the MCR of nmD101H rapidly decreased below 40 kcps within the first 30 min of measurements (Figure 5E), indicating that the number of particles in the suspension was too low to provide sufficient scattered photons for DLS analysis and therefore pointing towards a very fast aggregation/sedimentation, in line with the higher MW bands present in the gel shift assay (Figure 3) and in the western blot (Appendix A).

Overall, at a physiological temperature all GCAP3 variants seemed to be more aggregation-prone upon Ca^2+^-binding than in the presence of Mg^2+^ or no ions.

### 2.6. Cation- and Temperature-Dependent Secondary and Tertiary Structure of GCAP3 Variants

The effects of the ion-induced conformational changes on GCAP3 variants were assessed by far and near UV-CD spectroscopy, which provided information about the protein secondary and tertiary structures, respectively. Since significant differences in the oligomeric properties of mGCAP3 and nmGCAP3 variants were detected by DLS, the CD spectra of these forms were collected both at 25 °C and at 37 °C, to evaluate potential temperature-dependent conformational rearrangements.

The representative far UV CD spectra of all variants (Figure 6) resembled the typical spectra of all α-helix proteins, characterized by local minima at 208 and 222 nm. At 25 °C, Mg^2+^-binding resulted in a small loss of secondary structure for both nmWT and mWT (Δθ/θ = −5% and −4%, respectively, Table 1), with detectable alterations of the spectral shape observed only for mWT (Figure 6A, θ_222_/θ_208_ = 0.86 vs. 0.90). Similar to most Ca^2+^-sensor proteins, both variants exhibited an increase in ellipticity at 222 nm upon addition of Ca^2+^ (Figure 6A,B, nmWT Δθ/θ = 4%, mWT Δθ/θ = 10%), accompanied by a significant rearrangement of the secondary structure in the case of mWT, as shown by the significant differences in the spectral shape descriptor θ_222_/θ_208_, switching from 0.86 to 0.95 (Table 1).

Mg^2+^-binding to apo mWT GCAP3 resulted in a slight decrease in ellipticity at 222 nm (Δθ/θ = −5%, Table 1) also at 37 °C (Figure 6D), at odds with nmWT (Figure 6C), which did not show any differences. Ca^2+^-binding, on the other hand, again induced an increase in the intensity of the spectrum (nmWT Δθ/θ = 10%, mWT Δθ/θ = 13%, Table 1) as well as a shape variation (nmWT θ_222_/θ_208_ = 0.87, mWT θ_222_/θ_208_ = 0.91) in both WT forms.

Interestingly, cation binding to nmD101H (Figure 6D) decreased the intensity of the spectra (Δθ/θ = −3%, Table 1), with minor differences in the shape only in the presence of Ca^2+^.

Near UV-CD spectra collected at 25 °C (Figure 7A,B) showed that both mWT and nmWT GCAP3 are structured even in absence of ions. However, the presence of the post-translational modification was found to affect the sensitivity to Mg^2+^. Indeed, while nmWT displayed a significant increase in ellipticity in both Tyr and Trp bands, such increase was less pronounced for mWT and it involved only the Tyr region. A major conformational rearrangement of aromatic residues was exhibited by WT GCAP3 upon Ca^2+^-binding (Figure 7A,B), as shown by the increase in ellipticity in all three aromatic regions, although significantly more pronounced for mWT (Figure 7B).

Interestingly, temperature was found to affect the structure of both the apo forms of WT GCAP3, and it influenced their structural response to ion binding. Indeed, nmWT (Figure 7C) showed an overall more intense signal with a less defined, fine structure at 37 °C compared to that at 25 °C (Figure 7A), while the spectrum of apo mWT (Figure 7D) exhibited a loss of intensity and fine structure. Mg^2+^-binding resulted in an increase in intensity of the spectrum in all three regions for both nmWT and mWT, thus implying that mWT is more structurally sensitive to Mg^2+^ at 37 °C rather than at 25 °C. Interestingly, the conformational changes exhibited by both WT forms upon Ca^2+^-binding at 25 °C were completely absent at 37 °C, as shown by the substantial overlap of the spectra in the presence of Mg^2+^ and Ca^2+^ (Figure 7).

The disease-associated nmD101H variant, on the other hand, displayed a completely opposite behavior, with apo nmD101H being significantly more structured than its WT counterpart (Figure 7E) and the ion-bound forms displaying a decrease in ellipticity. Notably, the absence of changes in tertiary structure associated with Ca^2+^-binding following prior incubation with Mg^2+^ was a common feature, shared by all three GCAP3 variants.

The dynamic behavior of the oligomeric state of GCAP3 variants detected by DLS prompted us to investigate whether time-dependent changes in the protein secondary structure could be observed by CD spectroscopy. We therefore monitored over a time frame of 5 h the θ_222_/θ_208_ ratio at 25 °C and 37 °C. None of the tested cases exhibited major variations at either temperature (Appendix A), although a slight increase in θ_222_/θ_208_ could be noticed for apo-nmWT (Appendix A, top) and nmD101H variants (Appendix A).

Temperature was found to significantly affect the time-dependence of the secondary structure of both WT variants (Table 2), an exception made for Mg^2+^-bound nmWT, which displayed no differences in spectral shape but only higher data scattering at 37 °C, a clear temperature-dependent feature shared by all tested variants. In detail, nmWT exhibited higher θ_222_/θ_208_ values at 37 °C in the absence of ions (0.84 ± 0.02 vs. 0.91 ± 0.03, Table 2) and upon Ca^2+^ binding (0.84 ± 0.02 vs. 0.85 ± 0.03, Table 2), while mWT showed a different behavior, as the θ_222_/θ_208_ index increased at 37 °C in the apo form (0.83 ± 0.03 vs. 0.85 ± 0.05, Table 2) and decreased in the ion-bound form (0.86 ± 0.03 vs. 0.82 ± 0.05 in the presence of Mg^2+^ and 0.86 ± 0.03 vs. 0.83 ± 0.03 in the presence of Ca^2+^, Table 2). The nmD101H variant, on the other hand, showed statistically significant differences with the nmWT only when cation-bound, as shown by the increase in θ_222_/θ_208_ from 0.85 ± 0.04 to 0.88 ± 0.03 in the presence of Mg^2+^, and from 0.85 ± 0.03 to 0.91 ± 0.03 in the presence of Ca^2+^ (Table 2).

### 2.7. Analysis of Hydrophobic Surfaces in GCAP3 Variants by 1-Anilinonaphthalene-8-Sulfonic Acid (ANS) Fluorescence

To assess whether the unusual properties of GCAP3 variants relate to dynamic changes in the accessibility of protein hydrophobic surfaces upon cation binding, we recorded fluorescence emission spectra of the hydrophobic probe ANS and monitored the variation in hydrophobicity of GCAP3 variants over time at two different temperatures.

The fluorescence spectra collected at 25 °C (Figure 8A) suggested that the hydrophobicity of nmGCAP3 slightly decreases upon cation binding, moreover, no differences could be identified between the Mg^2+^-bound and the Ca^2+^-bound forms. On the other hand, mWT did not exhibit any appreciable variation in hydrophobicity upon cation binding (Figure 8B), although it is worth noticing that, overall, mWT seems significantly less hydrophobic than nmWT. Similar conclusions could be drawn by analyzing the spectra of GCAP3 variants at 37 °C (Figure 8C–E), although at the higher temperature, also nmWT showed almost overlapping spectra regardless of the ion-loading conditions (Figure 8C). Surprisingly, nmD101H exhibited an appreciable increase in fluorescence upon Mg^2+^ binding (Figure 8E), suggestive of an increased hydrophobicity, which was significantly more evident in the presence of Ca^2+^.

The time-evolution of ANS fluorescence over 110 min highlighted small differences in the dynamics of hydrophobic surfaces exposure at 25 °C and 37 °C for nmWT in the absence of ions and in the presence of Mg^2+^, with a monotonic increase in fluorescence that reached a plateau in approximately 60–70 min (Appendix A, top left and middle panels). A different trend could be observed in the presence of Ca^2+^, as the fluorescence emission reached a plateau after 70 min at 37 °C, while at 25 °C we identified a rising phase in the first 30 min followed by a decrease in the following 60 min, after which fluorescence reached a plateau at lower relative values.

A completely different behavior could be observed for mWT (Appendix A, lower panels) in all three conditions, where fluorescence intensity reached its peak within the first 30 min and then gradually decreased. The time-dependence of mWT’s hydrophobicity was more apparent than that of nmWT, as shown by the different trends shown by the apo and Mg^2+^-bound forms (Appendix A, bottom left and middle panels). The Ca^2+^-bound form of mWT showed a significantly anticipated decrease of ANS fluorescence at 25 °C (Appendix A, bottom right panel).

Interestingly, the D to H substitution of residue 101 of nmGCAP3 significantly altered the dynamics of hydrophobic exposure over time under all tested conditions. At odds with nmWT, apo nmD101H indeed displayed a mild decrease in hydrophobicity after 30 min, followed by a more substantial decrease over the last 20 min (Appendix A, left panel). Similarly, Mg^2+^-bound nmD101H presented a pronounced decline in hydrophobicity already after 60 min, whereas upon Ca^2+^ binding the ANS fluorescence monotonically decreased almost immediately.

### 2.8. MD Simulations Provide Atomistic Interpretation of the Effects of the D101H Substitution on nmGCAP3 Flexibility and Ca^2+^-Binding

To investigate at atomistic level the molecular consequences of the substitution of Asp101 with a His residue we ran exhaustive 2 µs all-atom MD simulations for both nmWT and nmD101H variants. The consistency and convergence of the two replicas was verified (Appendix A) by a combination of descriptors based on principal component analysis (PCA), as elucidated in the Methods.

We assessed the flexibility of the backbone of the two GCAP3 variants by means of the Root–Mean Square Fluctuation of Cα, which represents the time-averaged root–mean square displacement with respect to the average structure (Figure 9A). The analysis of the RMSF profiles highlighted that the backbone of nmD101H is overall almost 1.6-fold more flexible than nmWT, with remarkable variations in the exiting helix of EF1 (αF1), the Ca^2+^-binding loop of EF3 and the linker region connecting EF3 and EF4.

The visual inspection of the MD trajectories allowed us to evaluate the effects of the substitution of D101 (the first Ca^2+^-coordinating residue of EF3) with the non-conservative histidine on Ca^2+^-binding. The loss of the highly conserved Asp in position 1 of the Ca^2+^-binding loop of EF3 implied the lack of a carboxyl group necessary for maintaining the correct pentagonal bi-pyramidal geometry required for Ca^2+^-coordination. Moreover, the introduction of a His residue in a strongly negatively charged region caused a major distortion of the Ca^2+^-binding loop. Such distortion caused the progressive loss of Ca^2+^-coordination by the sidechain of N105 and the backbone carbonyl of S107 (Figure 9B), thus causing the ion to juggle among D103, E112 (and D109, to a lesser extent) until the bidentate E112 was too far to keep Ca^2+^ in the binding loop, ultimately leading to the dissociation of the ion from EF3 (Figure 9B, Appendix A).

## 3. Discussion

The biochemical and biophysical properties of GCAP1 and GCAP2 have been extensively investigated in previous studies, which highlighted species-dependent characteristics; however, although GCAP3 was the first GCAP whose three-dimensional structure was resolved by crystallography [7], very little is known about its function so far. No disease-causing variants have been identified in the *GUCA1C* gene, except for an interesting variant that was identified in this study as the sole possible cause in a RP patient and was therefore analyzed with respect to its effect on protein function. Indeed, genetic screening revealed a nucleotide substitution in *GUCA1C* corresponding to a novel missense mutation, p.D101H, in the encoded protein GCAP3. Due to its prominent role in Ca^2+^ coordination, residue D101/100 is highly conserved in GCAPs, and amino acid substitutions in the same position in GCAP1 are associated with autosomal cone dystrophy (D100E) [44] and cone-rod dystrophy (D100G) [28]. In principle, this may suggest a similar pathogenic mechanism which, however, was refuted by our investigation. The molecular properties of GCAP3 have indeed been found to be significantly different from those of its GCAP1 [48,49] and GCAP2 [13] paralogs studied under similar conditions.

Analytical SEC results showed that, in line with what was observed for GCAP1 [5,50] and GCAP2 [13], GCAP3 also forms dimers (Figure 4). Myristoylation and the addition of Ca^2+^ decreased the apparent hydrodynamic radius of WT GCAP3, while the D101H substitution led to the co-presence of higher MM complexes. The reduction in signal intensity observed for both nmWT and nmD101H suggests that the myristoylation of GCAP3 somehow triggers an aggregation process, which may lead to a significant loss of the effective amount of functional protein in solution. It should be noted that, while GCAP1 and GCAP2 expressed in E. coli under similar conditions can be efficiently myristoylated with an efficiency exceeding 90%, with GCAP3 the efficiency was significantly lower (~58% for WT and ~60% for D101H mutant). This phenomenon was previously noted by Haeseeler et al. [8] who measured a 75% efficiency of myristoylation when human GCAP3 was expressed in insect cells, therefore this seems to be an intrinsic property of GCAP3. Although Haeseleer et al. concluded that myristoylation was not absolutely required for GCAP3 activity, our assays showed that, in the absence of myristoylation, the activation of GC1 is indistinguishable from that in the absence of GCAP3 (Figure 3).

While essential for the target activation, myristoylation seems to exert a destabilizing effect on human GCAP3 in terms of colloidal and oligomeric properties, as proven by analytical SEC, DLS experiments, CD spectroscopy and ANS fluorescence, which overall point to a temperature- and time-dependent aggregation process that is particularly apparent in the presence of Ca^2+^ (Figure 5). This is a clear difference between GCAP3 and both GCAP1 [26] and GCAP2 [13]. There is another surprising peculiarity of GCAP3, which is apparently independent of the presence of myristoylation. Ca^2+^ ions indeed bind to the protein and trigger a conformational change even following the prior incubation with Mg^2+^ (Figure 3D), and the resulting secondary and tertiary structures show the typical features of an all α-helix protein (Figure 6 and Figure 7). Although the process is temperature-dependent, at both investigated temperatures each analyzed variant reached a relatively constant hallmark of secondary structure as shown by time-dependent CD spectroscopy (Appendix A).

Apart from the already mentioned difference observed in the putative role of GCAP3 myristoylation in GC activation, our results significantly differ from those of Haeseleer et al., who reported a similar activation profile of GC stimulated by human GCAP1 or GCAP3 [8]. While the IC_50_ value detected for GCAP3 in this study is compatible with the physiological change in [Ca^2+^] occurring during the phototransduction process (Figure 3C), the activation of GC1 by GCAP3 is approximately 6-fold lower compared to the activation induced by GCAP1 (Figure 3B). A possible explanation for the discrepancy could be that Haeseleer et al. performed experiments with either bovine GC1 or the washed rod outer segments from bovine retina, which contain both GC1 and GC2 isozymes. To our knowledge, experiments were not performed with human GC1. A species-dependent optimization of the GC-GCAP complexes has been recently hypothesized in light of experiments with reconstituted systems [51], therefore what has been observed for the activation of a bovine GC may not be extensible to the human ortholog.

The newly identified D101H mutation in GCAP3 leads to a correctly folded protein that retains some capability of Ca^2+^ binding (Figure 6 and Figure 7). However, the presence of Ca^2+^ rapidly drives protein precipitation, as shown by DLS (Figure 4) and time-dependent ANS fluorescence (Appendix A). MD simulations suggest that the point mutation induces a high structural flexibility throughout the protein sequence (Figure 9) and, fully in line with experimental observation, predicts that in approximately 700 ns, Ca^2+^ coordination in EF3 is completely lost, with a major rearrangement of the whole loop region that accommodates H100 in the position previously occupied by Ca^2+^; this distortion might constitute the trigger of the aggregation process. Regardless of the presence of myristoylation, the D101H was shown to be completely incapable of activating GC1, as shown by the cGMP production which was undistinguishable from that of the control in the absence of GCAP3 (Figure 2A). This is a major difference with the D100E [44] and D100G [28] variants of GCAP1, which drive retinal degeneration by constitutively activating GC1. Our data suggest that the novel D101H variant of GCAP3 is associated with retinitis pigmentosa, but the underlying pathogenetic mechanism currently remains unknown, as no similarity to GCAP1 could be postulated. How a protein that according to previous studies is specifically expressed in cones [9] could drive a rod dystrophy is a puzzling issue, which deserves dedicated future studies.

## 4. Materials and Methods

### 4.1. Patient Enrollment and Retrieval of Blood Samples

The patient in this study was recruited and clinically examined at the Eye Hospital, University of Tübingen, Germany. Genomic DNA was extracted from peripheral blood using standard protocols.

### 4.2. Clinical Evaluation

Ophthalmic examination included detailed medical history, best corrected visual acuity (BCVA) testing, kinetic perimetry (Octopus 900, Goldmann-III4e-Stimulus, Haag-Streit GmbH, Wedel, Germany), slit lamp examination, fundus examination and photography including fundus autofluorescence images, full-field electroretinography (ERG) according to the ISCEV (International Society for Electrophysiology of Vision) standards and optical coherence tomography (OCT; Spectralis^®^ OCT/HRA, 55 degrees, Heidelberg Engineering, Heidelberg, Germany).

### 4.3. Genetic Diagnostic Testing

Genetic diagnostic testing of the patient was performed by genome sequencing. Methodological details have already been published [41]. Genotyping of family members for the c.301G > C variant in *GUCA1C* was performed with bidirectional Sanger sequencing of PCR products obtained with forward primer *GUCA1C*-Ex2-f (5′-cacgaatgctgagttctcaaa-3′) and *GUCA1C*-Ex2-r (5′-cctactccataggaagggaaa-3′).

### 4.4. Cloning, Protein Expression and Purification

The synthetic gene of human GCAP3 (Uniprot entry: O95843) was cloned into a pET11-E6S plasmid. The D101H mutation was generated by PCR site-directed mutagenesis using KOD Hot Start Polymerase (Merck-Millipore, Darmstadt, Germany) followed by incubation at 30 °C for 1 h with unique Kinase-ligase-DpnI (KLD) enzyme-mix (New England Biolabs, Frakfurt a.M., Germany). The mutated DNA plasmid was amplified, and the point mutation verified by sequencing (Eurofins Genomics, Ebersberg, Germany). NmWT and nmD101H GCAP3 were heterologously expressed after transformation of plasmids in *E. coli* BL21 DE3 cells. MWT and mD101H GCAP3 were heterologously expressed in the same bacteria after co-transformation with the plasmid of GCAP3-WT or GCAP3-D101H and the plasmid of pBB131-yNMT encoding for N-myristoyl transferase enzyme, necessary to obtain the post translation modification. Briefly, cells were allowed to grow at 37 °C until absorbance at 600 nm reached 0.6, when protein expression was induced by adding 1 mM Isopropyl-β-d-1-thio-galactopyranoside (IPTG). For the myristoylated forms, 50 µg/mL in 50% ethanol (pH 7.5) was added at an absorbance of 0.4. After the induction, the bacteria were allowed to grow at 37 °C and the pellets were collected after 4 h. For each variant, the pellet was lysed, and the inclusion bodies resuspended in 6-M guanidinium hydrochloride for o.n. solubilization.

Proteins were refolded by dialysis and purified by anion exchange chromatography as the first step. For WT forms we used an HiPrep Q column (HP 16/10 GE Healthcare, now Cytiva, Freiburg im Breisgau, Germany) equilibrated in 20 mM Tris-HCl, 2 mM EGTA and 1 mM DTT pH 7.5 and then the proteins were eluted with a linear salt gradient from 0 to 1 M of NaCl. For mutant forms we used a Resource Q column (Cytiva, Freiburg im Breisgau, Germany) equilibrated in 20 mM Tris-HCl, 2 mM Ca^2+^ and 1-mM DTT pH 7.5 and then the proteins were eluted with a salt linear gradient from 0.02 to 0.55 M of NaCl. Fractions containing the expressed proteins were pooled, precipitated by ammonium sulfate, and further purified by size exclusion chromatography. A Sephacryl S-200 (HiPrep 26/60 HR, GE Healthcare) equilibrated with 20 mM Tris-HCl pH 7.5, 150 mM KCl and β-Mercaptoethanol was employed for WT proteins while for mutant proteins the Superdex 75 (HighLoad 26/60) column was equilibrated with 20 mM Tris-HCl pH 7.5, 150 mM NaCl, 2 mM CaCl_2_ and 1 mM DTT. Purity of protein samples was analyzed by SDS-PAGE, finally samples were aliquoted and flash-frozen in liquid nitrogen and stored at −80 °C or exchanged against 50 mM ammonium hydrogen carbonate, lyophilized and stored at −20 °C until use. Degree of myristoylation was 58.35% for the WT and 60.55% for the D101H variant, as checked by reversed phase high performance liquid chromatography (HPLC) using a Luna RP-C18 column (5 µm, Phenomenex, Aschaffenburg, Germany) and running a gradient from 0.1% trifluoracetic acid (TFA) in double distilled water to 0.1% TFA in 100% acetonitrile.

### 4.5. GC Activity Assay

Recombinant human GC1 was expressed in HEK293 cells following PEI-DNA transfection and the positive cell pool was selected by geneticin (500 µg/mL) [52]. GC activity was measured after reconstitution of cell membranes and GCAP3 variants. Briefly, the cells were incubated on ice for 30 min, lysed and centrifuged. The pellet was suspended in resuspension buffer (50 mM HEPES pH 7.4, 50 mM KCl, 20 mM NaCl, 1 mM DTT and 1:500 mPIC) and the concentration of membrane proteins was determined by a Bradford assay [53] with additional 40% n-Octyl-β-D-glucopyranoside (OGP).

Lyophilized GCAP1 and GCAP3 variants were resuspended in 10 mM HEPES/KOH (5 mM for GCAP1) pH 7.4 using an ultrasonic bath on ice for 5 min, then protein concentration was estimated by Bradford assay. Human GC1 membranes were incubated for 10 min with 10 µM GCAP3 variants or 5 µM GCAP1 in the presence of 1 mM free Mg^2+^ and low (<19 nM) or high (~30 µM) Ca^2+^ concentrations as previously described [1,38,39,54]

The production of cGMP was analyzed by reversed phase HPLC (VWR-Hitachi LaChrom Elite column), the area corresponding to the cGMP peaks was determined with the EZCChrome Elite 3.1.6 software; finally cGMP was quantified using a standard calibration curve. Data shown in Figure 3 were normalized to the duration of the assay and the total amount of protein.

The half-maximal activation of GC (IC_50_) was measured after 10 min incubation of GCAP3 variants and in membrane suspensions containing GC1. Increasing free Ca^2+^ concentration (<19 nM–1 mM) was adjusted by a Ca^2+^-EGTA buffer system [1,38,39,54]. GC activity was plotted as a function of the Ca^2+^ concentration and the average of three technical replicates was fit to a 4-parameter Hill sigmoid.

### 4.6. Gel Shift Assay and Western Blot

Cation-induced conformational changes were monitored on 15% acrylamide SDS-PAGE to evaluate differential electrophoretic mobility of GCAP3 variants under denaturing conditions. Proteins were resuspended in 50 mM Tris-HCl pH 8.0 at 30 µM concentration and incubated for 5 min at 25 °C in presence of 5 mM EDTA, 1 mM Mg^2+^ + 5 mM EGTA or 1 mM Mg^2+^ + 5 mM Ca^2+^.

To identify dimers or aggregates of GCAP3 variants and to discriminate those from potential impurities, the western blot technique was performed. GCAP3 variants were loaded on a 12% acrylamide SDS-PAGE after 10 min incubation with 1 mM Ca^2+^, 1 mM EGTA, 1 mM Ca^2+^ + 1 mM Mg^2+^ or 1 mM EGTA + 1 mM Mg^2+^. The proteins on the SDS-PAGE of gel shift assay were electro-transferred on a nitrocellulose-membranes (NC-membrane) equilibrated with transfer buffer (Towbin-buffer). Then the NC-membranes were washed with TBS, blocked with blocking solution (1% milk powder in 0.05% Tween-20 in TBS), and incubated with anti-GCAP3 primary antibody (rabbit anti-hGCAP3; Novus Biologicals, Bio-Techne GmbH, Wiesbaden, Germany) for 1 h (dilution 1:1000). Then, membranes were washed with 0.05% Tween-20 in TBS and incubated with the secondary antibody (goat anti-rabbit IgG; Jackson ImmunoResearch, Cambridgeshire, UK) for 1 h (dilution 1:10,000) and detected with a western blot substrate solution (Advansta, San Jose, CA, USA). The protein bands were acquired with BioRadV3 Chemidoc Imager.

### 4.7. Analytical Size Exclusion Chromatography (SEC)

Analytical SEC was performed using a Superose 12 10/300GL column (GE Healthcare) to estimate the protein apparent MM. The column was equilibrated with 20 mM Tris-HCl pH 7.5, 150 mM KCl, 1 mM DTT buffer and: (i) 500 µM EGTA, (ii) 500 µM EGTA and 1 mM Mg^2+^, or (iii) 1 mM Mg^2+^ and 500 µM Ca^2+^. Protein samples (~40 µM) were loaded and the elution profiles at λ = 280 nm were recorded at 25 °C. The resulting elution volume (*V_e_*) was then used to calculate the partition coefficient (*K_av_*) using the following equation (Equation (1)):(1)$$K_{av}=\;\left(V_{e}-V_{v}\right)\left(V_{t}-V_{v}\right)$$
where *V_v_* is the void volume (8 mL), and *V_t_* is the total column volume (25 mL).

The MM of proteins was then extrapolated from the calibration curve of log (MM) as a function of *K_av_* as described in ref. [55].

### 4.8. Dynamic Light Scattering (DLS) Measurements

The aggregation propensity of GCAP3 variants was monitored at 25 °C and 37 °C with a Zetasizer Nano-S (Malvern Instrument, Kassel, Germany) collecting the measurements for ~5h. GCAP3 variants were diluted to ~8 µM in 20 mM Tris-HCl pH 7.5, 150 mM KCl, 1 mM DTT buffer and: (i) 500 µM EGTA; (ii) 500 µM EGTA and 1 mM Mg^2+^; or (iii) 1 mM Mg^2+^ and 500 µM Ca^2+^. Protein samples were centrifuged at 4 °C for 15 min at 18,000× *g* and filtered with a Whatman Anotop 10 filter (cutoff 20 nm, Sigma-Aldrich, Taufkirchen, Germany) before the measurements. Each sample was equilibrated for 2 min at 25 °C or 37 °C and at least 120 measurements, each consisting of 14 or 16 repetitions, were collected approximately every 150 s for 5 h.

### 4.9. Circular Dichroism (CD) Spectroscopy

CD data were collected at 25 °C and 37 °C on a Peltier thermostated Jasco J-710 spectropolarimeter as previously described [26]. Briefly, GCAP3 variants were diluted in 20 mM Tris-HCl pH 7.5, 150 mM KCl, 1 mM DTT buffer and the spectrum of the buffer was recorded and subtracted to both near and far UV CD spectra. Near UV CD spectra of ~30 µM GCAP3 variants were recorded between 250 and 320 nm in a 1 cm quartz cuvette in presence of 500 µM EGTA and after sequential additions of 1 mM Mg^2+^ and 1 mM Ca^2+^, leading to approximately 500 µM free Ca^2+^. Far UV CD spectra of 8 µM GCAP3 variants were recorded between 200 and 250 nm in a 1 mm quartz cuvette, in the presence of 300 µM EGTA and after sequential additions of 1 mM Mg^2+^ and 600 µM Ca^2+^, leading to approximately 300 µM free Ca^2+^. To evaluate potential time-dependent conformational changes, far UV spectra were also collected every 5 min for 5 h in the same experimental conditions.

### 4.10. 1-Anilinonaphthalene-8-Sulfonic Acid (ANS)-Fluorescence Spectroscopy

Cation-dependent hydrophobicity changes in GCAP3 variants were analyzed by exploiting the differential emission of the hydrophobic probe 1-Anilinonaphthalene-8-Sulfonic Acid (ANS). All spectra were collected with a Jasco FP 750 spectrofluorometer at 25 °C and 37 °C in the 400–650 nm range after excitation at 380 nm with excitation and emission bandwidths of 5 nm; each spectrum was an average of three accumulations.

GCAP3 variants were diluted to a final concentration of 2 µM in 20 mM Tris-HCl pH 7.5, 150 mM KCl, 1 mM DTT buffer, incubated for 2 min with 30 µM ANS in the presence of 500 µM EGTA and after sequential additions of 1 mM Mg^2+^ and 1 mM Ca^2+^. To address potential time-dependent hydrophobicity, fluorescence spectra were acquired every 10 min for 2 h in the same experimental conditions and the maximal fluorescence intensity was monitored.

### 4.11. Modeling and Molecular Dynamics (MD) Simulations

The PDB entry 2GGZ [7] (resolution 3 Å) containing the three-dimensional structure of Ca^2+^-loaded human GCAP3 (residues 22–186) was prepared following the “protein preparation” pipeline implemented in the BioLuminate (v. 4.0.139) module of the Maestro (v. 12.5.139, Schroedinger) modeling environment. In detail, the Chemical Components Dictionary database was employed to assign bond orders between couples of atoms (including Ca^2+^ ions), H atoms were generated by BioLuminate and the protonation state of ionizable residues at pH 7.5 was assigned by PROPKA. Finally, in silico mutagenesis of variant D101H was obtained by selecting the most similar rotamer to the original Asp residue.

Two independent 1-µs replicas of all-atom MD simulations were run for each GCAP3 variant on GROMACS (v. 2020.6) simulation package [56] using CHARMM36m forcefield [57]. Proteins were simulated in a dodecahedral box, solvated and their net charge was neutralized with 150 mM NaCl, for a total system size of 38,322–38,332 atoms. Structures were subjected to energy minimization and equilibration, as previously detailed in [18]. The exhaustiveness and consistency of the two replicas was assessed by means of Principal Component Analysis based on the covariance matrix of Cα as previously described in [58]. The projection of the trajectories on the first two PC (2D-projections, extracted from the single replicas and the concatenated trajectories), representing the largest collective motions of the protein, were subjected to Linear Discriminant Analysis and Root-Mean Square Inner Product (RMSIP) of the first 20 PC (constituting the essential subspace) as detailed in previous work [59]. The flexibility of the backbone of GCAP3 variants was assessed by Root–Mean Square Fluctuation (RMSF) of Cα, which represents the time-averaged Root–Mean Square Deviation with respect to the average structure.

## Figures and Tables

**Figure 1 ijms-23-03240-f001:**
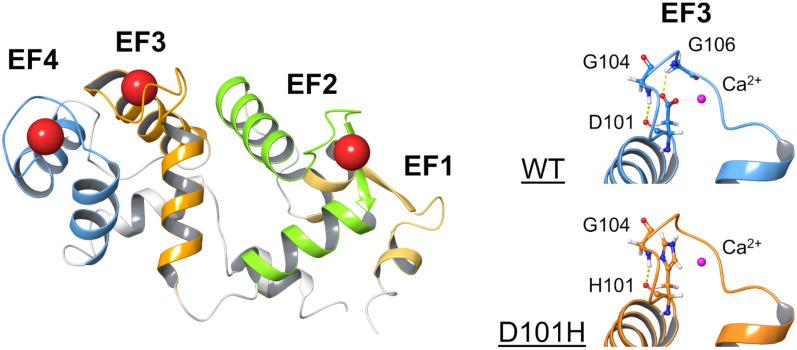
Structural model of D101H GCAP3 and clinical retinal imaging. Three-dimensional model (**left**) of non-myristoylated human GCAP3 in the Ca^2+^-loaded form (PDB entry: 2GGZ [7]). Protein structure is depicted as cartoon, with structural elements specifically colored (EF1 yellow, EF2 green, EF3 orange, EF4 blue) and Ca^2+^-ions represented as red spheres. Non-bonded interactions (**right**) involving D/H101 and nearby amino acids; residues are represented as sticks with H atoms in white; N atoms in blue and C atoms in red; Ca^2+^-ions are displayed as magenta spheres; H-bonds and electrostatic interactions are shown as yellow and magenta dashed lines, respectively.

**Figure 2 ijms-23-03240-f002:**
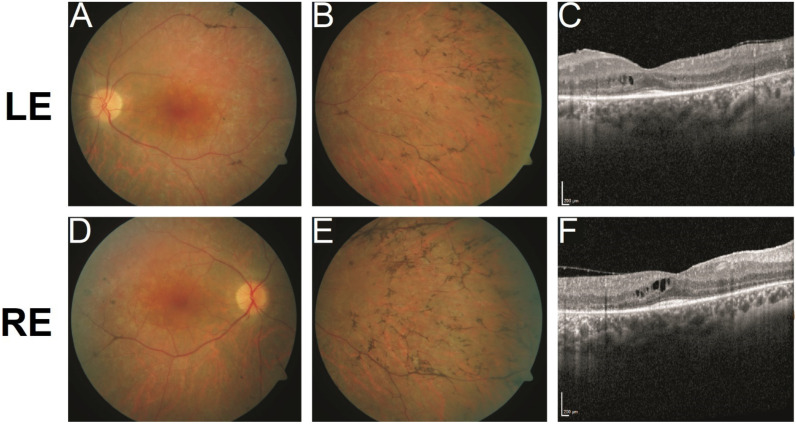
Central retinal fundus photography of (**A**) left eye (LE); and (**D**) right eye (RE). Mid-peripheral view of the fundus of (**B**) LE; and (**E**) RE. Vessel attenuation, atrophy of the retinal pigment epithelium and bone-spicule changes are evident in the fundus pictures, all typical for retinitis pigmentosa. Optical coherence tomography (OCT) of (**C**) LE; and (**F**) RE. Scale bars = 200 μm. Visualization of retinal layers with OCT shows reduced photoreceptor outer segments up to the fovea. Note several cystic spaces in the inner nuclear layer as a sign of macula edema, a common complication in retinitis pigmentosa.

**Figure 3 ijms-23-03240-f003:**
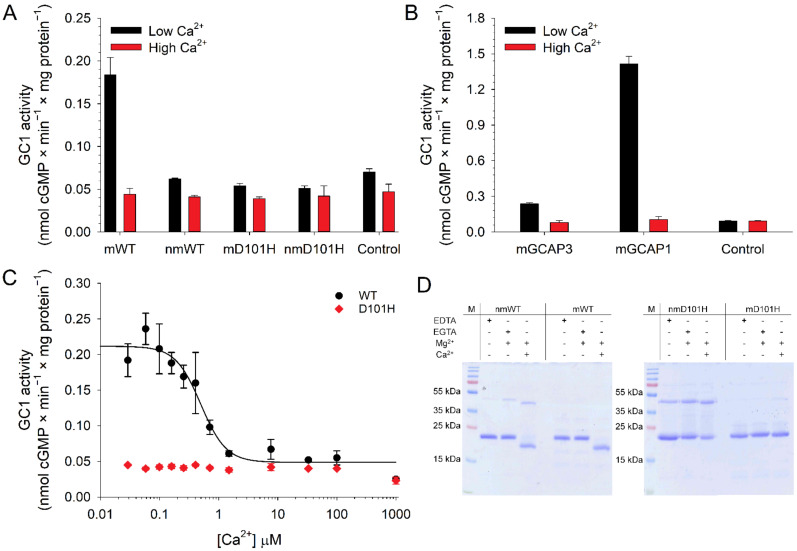
Regulation of human GC1’s activity and evaluation of Ca^2+^-sensitivity of GCAP3 variants. (**A**) Enzymatic regulation of human GC1 by 10 µM GCAP3 myristoylated (m-) and non-myristoylated (nm-) variants under activating (low Ca^2+^, <19 nM, black) and inhibiting conditions (high Ca^2+^, ~30 µM, red); (**B**) Enzymatic regulation of human GC1 by 10 µM mGCAP3 or 5 µM mGCAP1 under the same conditions. Control refers to the basal GC1 activity in the absence of regulators; (**C**) GC1 activity at different Ca^2+^ concentrations (<19 nM–1 mM) of 10 µM mWT (black) or mD101H (red) GCAP3; activation by mWT GCAP3 was half-maximal (IC_50_) at 0.52 µM; (**D**) 15% SDS-PAGE of ~7 µg m/nm WT and D101H GCAP3 upon incubation with 5 mM EDTA, 5 mM EGTA + 1 mM Mg^2+^ and 1 mM Mg^2+^ + 5 mM Ca^2+^.

**Figure 4 ijms-23-03240-f004:**
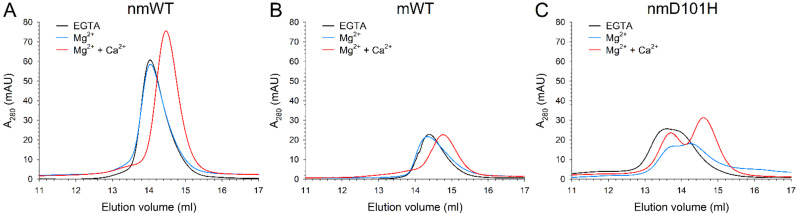
Analytical SEC chromatograms of ~40 µM (**A**) nmWT, (**B**) mWT and (**C**) nmD101H GCAP3 in the presence of 500 µM EGTA (black), 500 µM EGTA and 1 mM Mg^2+^ (blue), or 1 mM Mg^2+^ and 500 µM Ca^2+^ (red).

**Figure 5 ijms-23-03240-f005:**
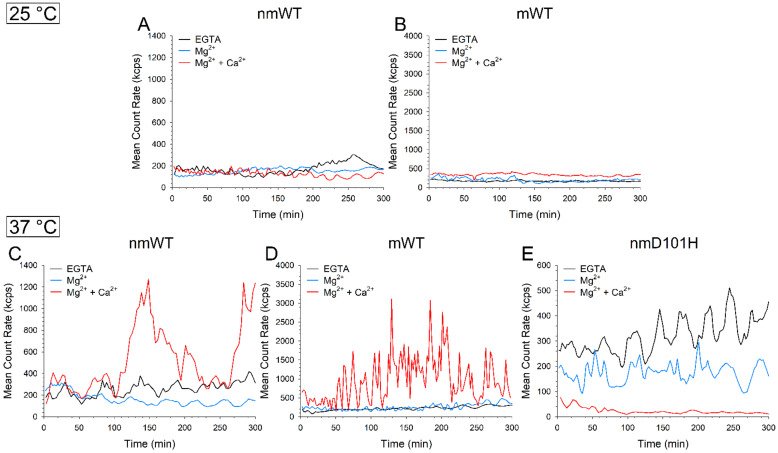
Time evolution over 5 h at 25 °C of the mean count rate of ~8 µM (**A**) nmWT; and (**B**) mWT GCAP3 in the presence of 500 µM EGTA (black), 500 µM EGTA and 1 mM Mg^2+^ (blue), or 1 mM Mg^2+^ and 500 µM Ca^2+^ (red). Time evolution over 5 h at 37 °C of the mean count rate of ~8 µM (**C**) nmWT; (**D**) mWT; and (**E**) nmD101H GCAP3 in the same experimental conditions as above.

**Figure 6 ijms-23-03240-f006:**
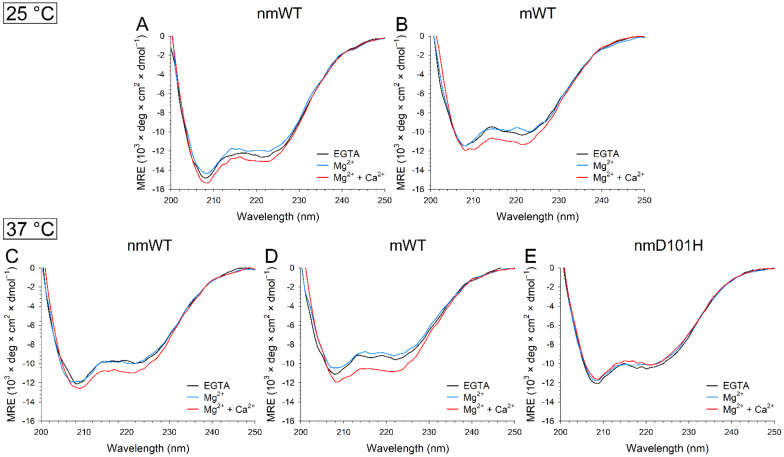
Representative far UV CD spectra recorded at 25 °C of ~8 µM (**A**) nmWT; and (**B**) mWT GCAP3 in the presence of 300 µM EGTA (black) and after sequential additions of 1 mM Mg^2+^ (blue) and 600 µM Ca^2+^ (red), leading to approximately 300 µM free Ca^2+^. Representative far UV CD spectra recorded at 37 °C of ~8 µM (**C**) nmWT; (**D**) mWT; and (**E**) nmD101H GCAP3 in the same experimental conditions as above.

**Figure 7 ijms-23-03240-f007:**
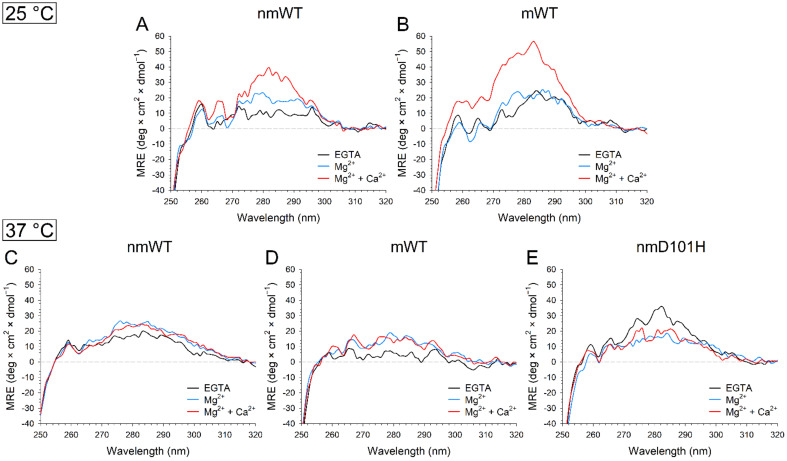
Representative near UV CD spectra recorded at 25 °C of ~30 µM (**A**) nmWT; and (**B**) mWT GCAP3 in the presence of 500 µM EGTA (black) and after sequential additions of 1 mM Mg^2+^ (blue) and 1 mM Ca^2+^ (red), leading to approximately 500 µM free Ca^2+^. Representative far UV CD spectra recorded at 37 °C of ~8 µM (**C**) nmWT; (**D**) mWT; and (**E**) nmD101H GCAP3 in the same experimental conditions as above.

**Figure 8 ijms-23-03240-f008:**
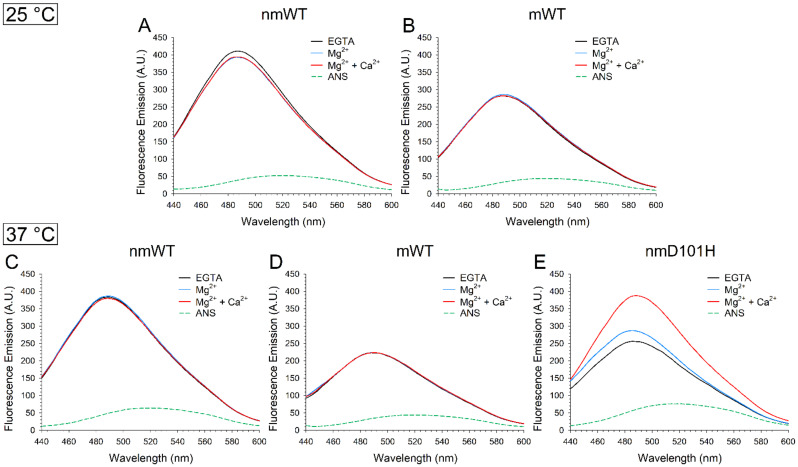
Fluorescence spectra of 30 µM ANS and 2 µM nmWT, mWT and nmD101H GCAP3 in the presence of 500 µM EGTA (black) and after sequential additions of 1 mM Mg^2+^ (blue) and 1 mM Ca^2+^ (red), leading to approximately 500 µM free Ca^2+^. Data were collected at 25 °C (upper panels) and 37 °C (lower panels). Fluorescence spectra recorded at 25 °C of 30 µM ANS and ~2 µM (**A**) nmWT; and (**B**) mWT GCAP3 in the presence of 500 µM EGTA (black) and after sequential additions of 1 mM Mg^2+^ (blue) and 1 mM Ca^2+^ (red), leading to approximately 500 µM free Ca^2+^. Fluorescence spectra spectra recorded at 37 °C of 30 µM ANS and ~2 µM (**C**) nmWT; (**D**) mWT; and (**E**) nmD101H GCAP3 in the same experimental conditions as above.

**Figure 9 ijms-23-03240-f009:**
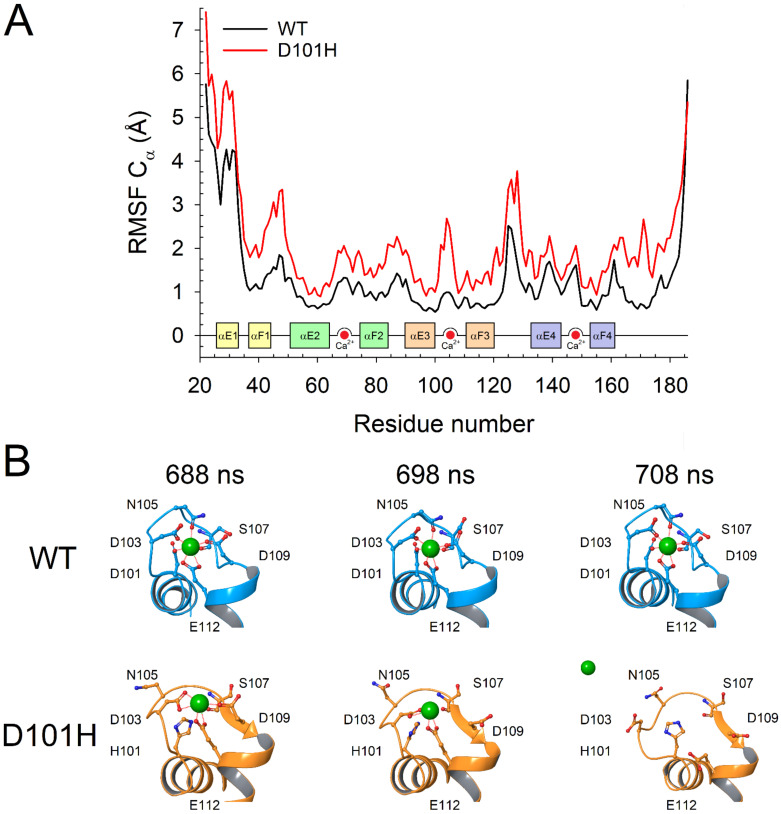
(**A**) Root–Mean Square Fluctuation (top panel) of Cα atoms calculated over 2 µs molecular dynamics (MD) simulations of Ca^2+^-loaded nmWT (black) and nmD101H (red) GCAP3. Inset shows secondary structure elements colored according to Figure 1, Ca^2+^-ions are represented as red circles. Snapshots of the Ca^2+^-binding EF3 motif (**B**) from 2 µs MD simulations of nmWT (blue) and nmD101H (orange) GCAP at three different time-frames; protein structure is shown in cartoon, Ca^2+^ ion is represented as a red sphere, Ca^2+^-coordinating residues D/H101, D103, N107, S107, D109 and E112 are labelled and depicted as sticks with N atoms in blue and O atoms in red, zero-order bonds with the Ca^2+^-ion are shown as dashed red lines.

**Table 1 ijms-23-03240-t001:** Results from Analytical SEC and CD spectroscopy.

			25 °C		37 °C	
Variant	Condition	MM (kDa) ^a^	θ_222_/θ_208_ ^b^	Δθ/θ ^c^	θ_222_/θ_208_ ^b^	Δθ/θ ^c^
	**Apo**	54.2	0.85	-	0.83	-
**nmWT**	**Mg^2+^**	54.0	0.84	−0.05	0.84	0.0
	**Ca^2+^**	41.2	0.85	0.04	0.87	0.1
	**Apo**	42.9	0.90		0.86	
**mWT**	**Mg^2+^**	44.1	0.86	−0.04	0.88	−0.05
	**Ca^2+^**	33.8	0.95	0.10	0.91	0.13
	**Apo**	69.3			0.86	-
**nmD101H**	**Mg^2+^**	46.8 (72.3)			0.86	−0.03
	**Ca^2+^**	37.6 (66.9)			0.87	−0.03

^a^ Estimated molecular mass (MM) of the main peak, when present; the estimated MM of the second peak is reported in brackets; ^b^ ratio between the ellipticity value recorded at 222 and 208 nm; ^c^ the relative ellipticity change was calculated as (θ_222_^ion^–θ_222_^apo^)/θ_222_^apo^.

**Table 2 ijms-23-03240-t002:** Time evolution over 5 h of the ratio between the ellipticity at 222 nm and 208 nm of GCAP3 variants in the presence of 300 µM EGTA; 300 µM EGTA + 1 mM Mg^2+^; 300 µM Ca^2+^. Data collected at 25 °C and 37 °C is reported as average ± standard deviation.

	nmWT		mWT		nmD101H
	25 °C	37 °C	25 °C	37 °C	37 °C
**EGTA**	0.84 ± 0.02	0.91 ± 0.03 ^a^	0.83 ± 0.03	0.85 ± 0.05 ^a^	0.90 ± 0.03
**Mg^2+^**	0.85 ± 0.03	0.85 ± 0.04	0.86 ± 0.03	0.82 ± 0.05 ^a^	0.88 ± 0.03 ^b^
**Ca^2+^**	0.84 ± 0.02	0.85 ± 0.03 ^a^	0.86 ± 0.03	0.83 ± 0.03 ^a^	0.91 ± 0.03 ^b^

^a^ difference between values at 25 °C and 37 °C are statistically significant, *p*-value 0.05; ^b^ difference with nmWT at 37 °C is statistically significant, *p*-value 0.05.

## Data Availability

Data are not publicly available due to their size; however, they are available upon request from the corresponding authors.

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
