# Peer review of "Molecular Properties of Human Guanylate Cyclase-Activating Protein 3 (GCAP3) and Its Possible Association with Retinitis Pigmentosa"

_ijms, 2022, doi:10.3390/ijms23063240_

Round 1

Reviewer 1 Report

Author show in this that GCAP3 can activate GC1 but with lower rate in comparison to GCAP1. They also showed that myristolation is needed for the activity and that point mutated protein also cannot activate GC1. GCAP3 was shown to have significantly different behavior in multiple different assays, but some of the obtained results seemed also confusing. With some minor problems in other parts, especially the aggregation tendency and thermal stability are not easily in line and must be explained. Below, some points are listed in more details.

  1. The manuscript follows the same pattern, making the same assays and tests than previous articles related to GCAP1 and GCAP2. However, as GCAP3 seems not to produce expected results, it is difficult to understand why analysis is left to same level. Especially thermal stability and aggregation data seems not to match. Author have used only ellipticity for Tm measurements, even they have used ANS for other things. ANS would serve a control assay for Tm also, and in addition, GCAP1 could have been used as a control. With these controls, it could have been judged if Tm is correct or is there some artefacts from purification/refolding procedures or something else. Current results partly point towards aggregation (low stability) and partly high stability without a tendency to aggregate. Which ever is true, might have effect on activity also.
  2. Unlike issues related to stability, all the other thing are just very minor. One of these is that figures and figure legends could be improved. For example Figure 1 could be divided in two (A and the rest), so that the legend from B onwards could be reorganized and improved. Currently it is quite hard. Also it would be clearer to use letters in all figures (figures 3-8). With this and with additional details in the figure legend, understanding the point of each figure without a main text could be possible. Also terms and abbreviations should be made clear, and abbreviations should be opened in the main text. E.g. in the figure 2, WT, mWT and nmWT terminology is used. Similarly there is a name control, which is not explained. Also by improving titles, like 2.2, to make to more informative, would improve the manuscript.

All in all the text is quite clear and conclusions are mainly justified. After clarifying the issues related to aggregation and stability, the story will be solid.

Reviewer 2 Report

Guanylate cyclase-activating proteins (GCAPs) are neuronal calcium sensors involved in the regulation of the membrane-bound guanylate cyclases via a Ca2+-mediated feedback mechanism that is essential to achieve a fine modulation of the photo-transduction cascade in photoreceptors.  The authors identified a subject carrying a point mutation in GUCA1C, encoding GCAP3, in heterozygous state, whose clinical phenotype is consistent with retinitis pigmentosa, thus suggesting the involvement of GCAP3 in inherited retinal disease.  However, the molecular mechanisms underlying GCAP3 function remain largely unexplored.  Thus, the aim of this study was to characterize the biochemical and biophysical properties of human GCAP3.  The authors found that myristoylated GCAP3 can activate guanylate cyclase with a similar Ca2+-dependent profile.  The non-myristoylated form did not induce appreciable regulation of guanylate cyclase, nor did the GCAP3 variant.  GCAP3 forms temperature-dependent aggregates driven by hydrophobic interactions.  The peculiar properties of GCAP3 were confirmed by molecular dynamics simulations which for the variant highlighted a very high structural flexibility and a clear tendency to lose the binding of a Ca2+ ion to EF-hand motif 3.  The authors conclude that GCAP3 has unusual biochemical properties and the point mutation resulting in a substantially unfunctional protein could trigger retinitis pigmentosa.  I did not have any major concerns, only several minor issues listed below:

Page 5 lines 173–175, “As shown in Figure 2A, only mGCAP3 was able to regulate GC1 in a Ca2+-dependent manner; indeed, in the presence of nmGCAP3 and D101H variants GC1 exhibited an activity comparable to that of the control, indicating a lack of enzyme regulation.”, Please define “mGCAP3” and “nmGCAP3” here.

Page 6 lines 219–221, “The elution profiles of nmWT and mWT GCAP3 in the absence of ions and in presence of Mg2+ (Figure 4) showed almost overlapping peaks, corresponding to a MM of ~54 and ~43 kDa, respectively, compatible with the dimeric form of the proteins.”, I guess Figure 3.

Page 10 line 340, “chaanges”

Page 13 lines 429–431, “To investigate at atomistic level the molecular consequences of the substitution of Asp101 with a His residue we ran exhaustive 2 μs all-atom MD simulations for both nmWT and nmD101H variants.”, Please define “MD”.

Round 2

Reviewer 1 Report

-